# How Semi-Urbanisation Drives Expansion of Rural Construction Land in China: A Rural-Urban Interaction Perspective

Yang Wang [ID], Chengchao Zuo *[ID] and Mengke Zhu

College of Public Administration, Huazhong Agricultural University, Wuhan 430070, China; wangyang9459@163.com (Y.W.); zhumengke@webmail.hzau.edu.cn (M.Z.)
* Correspondence: c.zuo@mail.hzau.edu.cn; Tel.: +86-(0)-15527861671

**Abstract:** The expansion of rural construction land in China has led to ecological consequences under the context of rural depopulation, despite government land use controls. While previous studies have investigated the local factors contributing to the expansion of rural construction land, the semi-urbanisation of urban immigrants distant from rural areas has received less attention. To better understand the connections between the semi-urbanisation in the urban areas and the construction land expansion in rural areas, this study constructed a network/spatial lag of N/X (N/SLX) model that incorporated the network lagged term of the socio-economic traits of semi-urbanised migrants to analyse how urban semi-urbanisation influenced rural construction land. Our findings suggest that both the income of rural-urban migrants and the difficulty of obtaining urban household registration are positively correlated with the expanding extent of rural construction land. Conversely, the living expenses of migrants and city economic development are negatively correlated with that of rural construction land. Considering our findings, we propose that policies facilitating the settlement and integration of rural out-migrants into cities and proceeding urbanisation based on county towns are crucial to curb the inefficient expansion of rural construction land.

**Keywords:** semi-urbanisation; rural construction land expansion; spatial analysis; network/spatial lag of X model





## 1. Introduction

The expanding footprint of construction land is commonly linked to the depletion of ecological land, thereby leading to diminished biodiversity and ecosystem degradation and posing a threat to the attainment of sustainable development goals [1,2]. In China, the rural construction land expanded from $1.25 \times 10^5$ km² in 2000 to $1.44 \times 10^5$ km² in 2020, thus surpassing the area of urban construction land in 2020 by about 2.5 times [3,4]. Concurrently, China has witnessed unprecedented urbanisation since the 1990s, which has been marked by a substantial migration of the rural population to urban areas for living and employment, thus resulting in a decline in the overall rural population [5,6]. Government statistics indicate that China's rural population decreased significantly by 27.80%, thus dropping from 807.39 million to 509.78 million between 2000 and 2020 [7,8]. At the prefecture level, the changes in rural population and rural construction land are mostly consistent with the national trend during the same period (Figure 1A,B). Moreover, about 93% of China's prefecture's per capita rural construction land area increased considerably (Figure 1C,D).

The substantial migration of people from rural to urban areas is a common phenomenon during the urbanisation process, which is typically accompanied by urban expansion, depopulation in rural areas, and reduced demand for rural construction land [9,10]. Albeit there are strict land use control measures imposed by the Chinese government, the area of rural construction land has still expanded and caused a series of issues in China, including inefficient land use and the loss of arable and ecological land [11,12]. Identifying

the mechanisms behind rural construction land expansion in China is, therefore, vital for the development of solution-focused policy instruments [13].

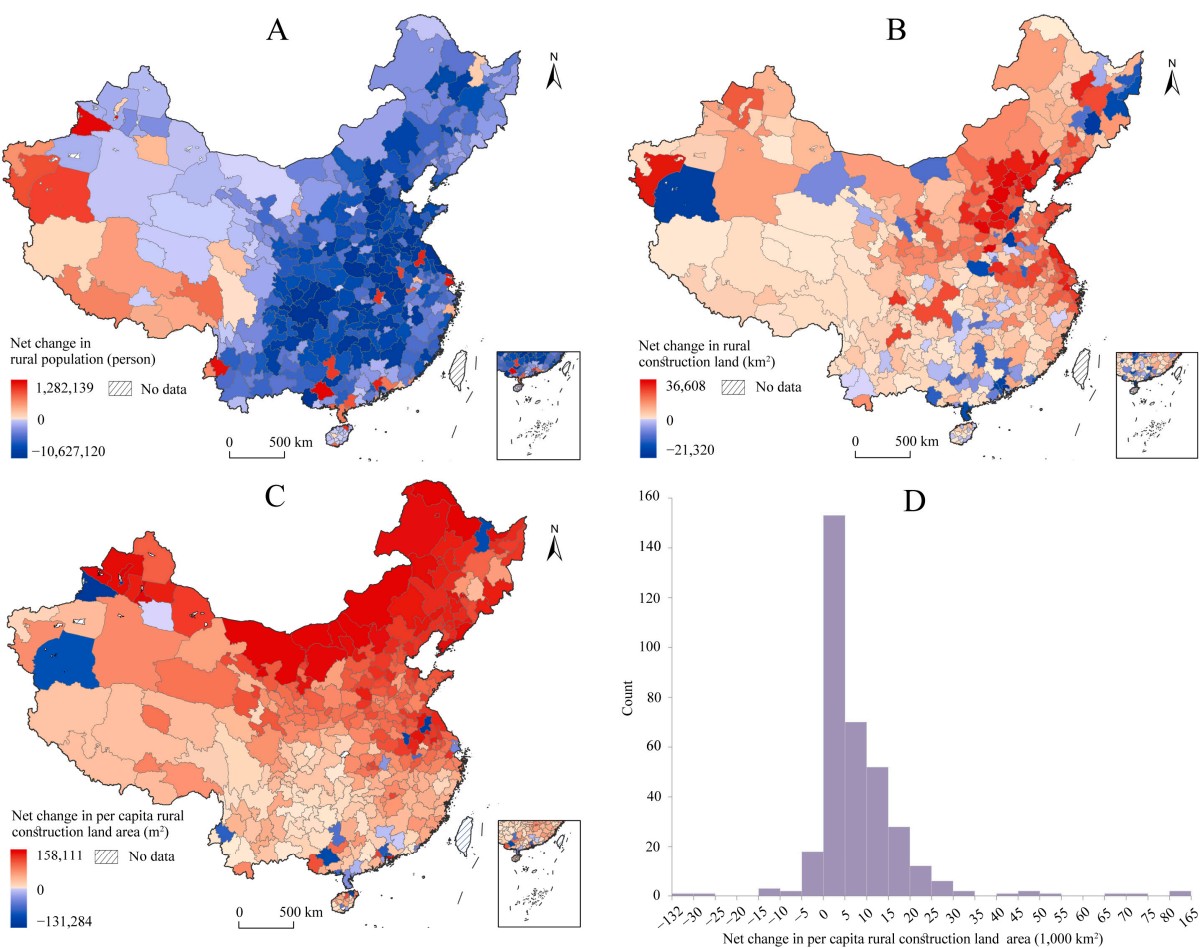

**Figure 1.** (**A**) Net change in rural population: 2000–2020. (**B**) Net change in rural construction land: 2000–2020. (**C**) Net change in per capita rural construction land area: 2000–2020. (**D**) Frequency distribution histogram of net change in per capita rural construction land area: 2000–2020.

Several studies have investigated the factors contributing to the expanding area of rural construction land [14–17]. Generally, topographic characteristics, permissive planning regulations, and the growth of rural populations have been widely acknowledged as global reasons for such expansion [14,18,19]. Evidence from developed countries has indicated the transition of affluent social groups from urban or suburban lifestyles to rural living, enticed by the charm of rural amenities [16,20], while in many developing countries, the expansion of rural construction land is often attributed to urban industrial transformation, economic development, and demographic shifts occurring in rural areas [2,21–23]. Ma et al. (2018) [24] and Liu et al. (2022) [25] discovered that construction land in the rural-urban fringe was growing in size because of industrial expansion in urban areas. Additionally, rural economic development provides greater financial resources for rural households to extend their homes, thus further contributing to rural construction land expansion [19]. Tan and Li (2013) [26] proposed that the increasing number of smaller households may be a factor behind rural construction land expansion in China. While most empirical studies have concentrated on local rural area features, such as the natural or socio-economic characteristics specific to each country [27], few have explored the consequences of rural outmigration on changes in rural land use. [28–31]. Liu et al. (2010) [32] investigated the driving mechanism of hollowing villages, which are characterised by vacant and unused rural housing located in the heart of these communities. The authors concluded that

farmers' migration for off-farm employment significantly contributed to the process. Shi and Wang (2021) [33] noted a positive correlation between rural outmigration and the extent of rural construction land in China. But despite the valuable insights these studies offer, the phenomenon of rural construction land expansion under circumstances where demand is decreasing (e.g., in China) has yet to be explored explicitly [25].

In contrast with many other countries, rural-urban migration in China takes place at an individual rather than a family level during the urbanisation process [6,34]. In this phenomenon, young individuals move to cities as *migrant workers* while leaving their family members behind in their home villages [35]. This unique form of incomplete urbanisation of the population is often referred to as semi-urbanisation, thereby signifying complex and substantial interactions between rural and urban areas [36,37]. To tackle the challenges posed by these distant causal relationships in land use changes, the telecoupling land system framework has been proposed since the 2010s [38,39]. Telecoupling land systems can be driven by various factors, including globalisation, migration, trade, and technological advances [28,40,41]. For example, Seto et al. (2012) [42] introduced a conceptual framework for urban land teleconnections, which captures the reciprocal changes between urban and rural areas. A series of studies have identified remittance receipt and labour force loss as pivotal factors in forest transitions in Latin America [40,43].

Based on the findings of existing studies, this research endeavours to address a crucial research question related to the influence of semi-urbanisation on changes in rural land use. In particular, it explores the impact of semi-urbanised rural-urban migration on the expansion of rural construction land. This study firstly developed a theoretical framework to elaborate the influence pathways of semi-urbanised migrants on the expansion of rural construction land from the perspective of rural-urban interaction. According to the mechanism, it designed a serious of variables to quantify the traits of the semi-urbanised migrants in each city—that is, their settlement expectations and socio-economic features. It subsequently constructed a network/spatial lag of X model to confirm how semi-urbanised migrants affect the extent of rural construction land through the rural-urban interactions embedded within migration networks. The methodology involved the design of network-lagged variables, which were represented as the product of the migration network weight matrix and the explanatory variable vectors associated with the features of semi-urbanised migrants. The methodology concerning the creation of the migration network/spatial weight matrix was estimated from the Weibo mobility index and the Seventh National Population Census of China, thus illustrating the interactions between rural and distant urban areas. The model enabled us to assess whether semi-urbanised migrants had a significant impact on the expanding area of rural construction land.

The present study makes a three-fold contribution to the existing literature. Firstly, it extends beyond prior research focused solely on local factors in rural and adjacent urban areas when examining the drivers of rural construction land expansion. By considering the effects of semi-urbanisation over distances, this study provides valuable insights for the development of policies aimed at addressing uncontrolled rural construction land expansion and subsequent environmental damage. Secondly, in comparison to previous vague explanations regarding the reasons for rural construction land expansion in rural-urban migration, this study offers a more explicit understanding by emphasizing the influence of settlement expectations and socio-economic features of semi-urbanised migrants. Thirdly, it enhances existing spatial empirical research by introducing a flow network dependence matrix, thereby expanding the scope of spatial dependence beyond distance or adjacency relationships.

## 2. Materials and Methods

### 2.1. Theoretical Analysis and Hypotheses

Based on the telecoupling land system framework, this paper developed a new theoretical framework to understand the impact of semi-urbanisation on the expansion of rural construction land over distances. Semi-urbanisation is characterised by the incomplete

integration of rural-urban migrations into the city and complicated urban–rural interactions [37]. We deconstructed semi-urbanisation into two dimensions and put it into the migration network of rural-urban interactions to elaborate. First, the remittances of semi-urbanised migrants from urban to rural areas shows that migrants still keep close relationships with their families in rural areas [40,44,45]. Second, the settlement expectations of semi-urbanised migrants in the city where they work and live present their integration into city life [46,47]. Low settlement expectations in the cities of semi-urbanised migrants generally imply their low degree of integration in the city and, thus, strong intentions of returning to their home villages [48,49].

Based on the above deconstruction of semi-urbanisation, we move on to the mechanism between semi-urbanisation and rural construction land expansion from a rural-urban interaction perspective. Unlike fully urbanised migrants, many semi-urbanised migrants still choose to maintain a rural resident identity (at least psychologically) [27,36] and retain their homestead in the rural area with the aim of returning to it at any point. This has created a decoupling relationship between the rural residents and rural construction land in China [33,50]. Furthermore, the relatively lower socio-economic status of semi-urbanised migrants might lead to over-investment in the rural housing market and the expanding area of rural construction land. On the one hand, these new migrants who have recently moved from rural areas to cities can earn higher incomes than they would have by farming [51], thus enabling them to invest more in their houses back home than those who remain [52,53]. On the other hand, higher living costs in cities have lessened the remittances that migrants can send to their home villages [54]; hence, there is a possibility that the expansion of rural construction land has become restrained.

The settlement expectations of semi-urbanised migrants also have an impact on the extent of rural construction land [48], especially the inclusive policies and economic development of cities. The dual-track household registration system—for instance, the Hukou system, where households are registered as either rural or urban, and the opportunity to transfer between the two is limited—solidifies the aforementioned urban-rural interactions [46,55]. In many cities, social benefits are bound to the location of the registered household; thus, local authorities tend to raise the household registration threshold to curb lower-income migrants from registering their households in the city to balance the public budget [56]. Such discriminatory policies further consolidate the phenomenon of semi-urbanisation by undermining rural migrants' expectations that they will integrate into the city. By contrast, the higher the cities' level of economic development (and, therefore, more employment opportunities and better welfare), the greater the willingness of new migrants to settle [46], thereby controlling the expanding area of rural construction land.

The aforementioned observations indicate that the increase in rural construction land cannot be viewed as an isolated phenomenon within rural areas. It is interconnected with the actions of semi-urbanised migrants residing and working in urban regions through the rural-urban migration networks across distant spaces. Consequently, it is important to place the rural construction land expansion mechanism within a narrative of urban-rural interaction. From this distinctive perspective, we hypothesised the following:

**Hypothesis 1.** *Semi-urbanisation would influence the extent of rural construction land through the settlement expectations of semi-urbanised migrants and their remittances from urban to rural areas.*

To uncover the mechanism more concretely and explicitly, we further specified Hypothesis 1 as the following:

**Hypothesis 2.** *Higher income levels of semi-urbanised migrants or household registration thresholds in cities where migrants live may lead to more expansion of rural construction land through the rural-urban interactions.*

**Hypothesis 3.** *Higher costs of living of semi-urbanised migrants or level of economic development in cities might restrain the expansion of rural construction land via rural-urban interactions.*

*2.2. Data and Variables*

2.2.1. Data Sources

Considering the data availability, this study selected 283 prefecture-level areas and 4 municipalities in China as research units. Prefecture-level rural construction land areas in 2018 were aggregated from 30 m × 30 m resolution raster land use data, which were derived from the Chinese Multi-Period Land Use Remote Sensing Monitoring Dataset (CNLUCC) published by the Resource and Environmental Science Data Registration and Publication System [4]. The 2018 Weibo mobility index was obtained from the China Data Lab Dataverse of Harvard Dataverse [57]. This dataset comprises 9.95 million geo-tagged posts generated by a cohort of 447,000 active users. The prefecture-level urban immigration was generated from the Seventh National Population Census of China. The prefecture-level average income and expenditure of migrants from rural to urban areas came from the 2017 CMDS (the National Health Commission, 2017), as the 2018 CMDS did not include information on immigrants' income and expenses. The Hukou registration index was jointly released by the China Centre for Behavioural Economics and Finance and the Survey and Research Centre for China Household Finance; the latest research is from 2016. The China Centre for Behavioural Economics and Finance is part of the Research Institute of Economics and Management of the Southwestern University of Finance and Economics [58]. To keep consistency with 2017 CMDS data, all control variables chose 2017. The number of households, production of secondary and tertiary industry, average slope, and agricultural output (total production of agriculture, forestry, husbandry, and fisheries) in each prefecture were collected from the corresponding *China City Statistic Book 2017* and the *Municipal Statistical Yearbook 2017*. The average slope was derived from the digital elevation model (DEM) provided by the Geospatial Data Cloud (http://www.gscloud.cn) (accessed on 23 October 2022).

2.2.2. Variables Selection and Descriptive Statistics

We set the area of rural construction land ($^{RC}Area$) in each prefecture in 2018 as a dependent variable $Y$. While controlling for local socio-economic and geo-morphological characteristics in rural areas (such as household numbers, average slopes, etc.), the expansion of rural construction land holds implications for the extent of rural construction land expansion [23].

According to our theoretical analysis, relatively low expectations of settling down and the remittances of semi-urbanised migrants are the core explanatory variables that drive rural construction land expansion. However, these factors are difficult to measure directly in practice; thus, we introduced a series of prefecture-level observation variables (including the average income and living expenditure of urban migrants, urban economic development, and discriminatory policies) to infer the attitude and activities of the cohort under study [36,46,53,59]. The average income of migrants from rural to urban areas was one of the primary explanatory variables, because their higher incomes allowed them to invest more in housing in their home villages [32,59]. The average income of rural-urban migrants in each city was estimated based on the individual-level sample data extracted from the 2017 China Migrants Dynamic Survey (CMDS; National Health Commission, 2017). The 2017 CMDS comprised 169,989 survey respondents located in 1290 county-level administrative regions in China. As the CMDS covers urban migrants from both urban and rural areas, we picked 140,563 samples whose household registration status was either agricultural or had just changed to urban residents and then calculated the average per-month income of these migrants by prefecture based on the selected samples.

Higher living expenditure in cities may influence new migrants' expectations of settling down and encourage them to invest more in their homesteads [46,47]. At the same time, the same expenditure restricts their ability to do so. In the present case, the average living expenditure of each prefecture was estimated based on the 140,563 samples of rural-urban migrants that were also used to reckon the average income of rural-urban

migrants. After working out the average per-month living expenditure of these migrants by prefecture, the average living expenditure of each prefecture was acquired.

The household registration threshold was measured using the Hukou registration index [58] to quantify the stringency of household registration restrictions in each city. Since the latest index measured by Zhang et al. [58] only went to 2016, we chose 2016. The Hukou registration index is a measure of the difficulty migrants face in obtaining an urban hukou from four channels (i.e., investments, home purchases, the talent programme, and employment). It has been widely interpreted as a hindrance to labour mobility, thus signalling a deficiency in local social acceptance and discrimination against migrants from rural areas. Given the limited educational and skill levels of rural migrants, we adopted the employment dimension of the Hukou registration index to gauge settlement expectations [49]. We expected that the lower the settlement expectation, the higher the index.

We used the production of secondary and tertiary industries in 2017 to measure the economic development of urban areas in each prefecture. Urban economies offer new migrants enhanced job prospects and improved access to public services, including healthcare and education. This, in turn, boosts their willingness to settle and reduces the likelihood of them expanding their rural homesteads [46,47].

To investigate the local socio-economic and environmental factors that might have influenced the extension of rural construction land, we followed Liu et al. [32] and Qu et al. [22]. In the context of Chinese land use policy, rural construction land allocation was determined on a household basis. The 2017 number of households was, therefore, considered as a major control variable. Because rural construction land on the plains is generally greater than that in hilly mountainous regions, we used average slope as a control variable. We used 2017 agricultural output (total production of agriculture, forestry, husbandry, and fisheries) to control the socio-economic development of the rural areas in each prefecture. Table 1 lists the variables utilised in this study.

**Table 1.** Descriptive statics of the variables.

| Variables | Symbol | Obs. | Min. | Max. | M | SD |
|---|---|---|---|---|---|---|
| Area of rural construction land (km$^2$) | $^{R}CL_i$ | 287 | 10.74 | 2165.18 | 463.91 | 466.22 |
| Income level of rural-urban migrants (RMB yuan) | $^{m}Inc_j$ | 287 | 1309 | 6366 | 3767 | 758.24 |
| Expenditure level of rural-urban migrants (RMB yuan) | $^{m}Exp_j$ | 287 | 1443 | 6726 | 3328 | 692.26 |
| Hukou registration index | $HRI_j$ | 287 | 0.00 | 1.51 | 0.30 | 0.25 |
| Production of secondary and tertiary industry (million RMB yuan) | $STI_j$ | 287 | 3361.05 | 3,052,271.12 | 176,834.08 | 374,876.35 |
| Total production of agriculture, forestry, husbandry, and fisheries (million RMB yuan) | $AFH_i$ | 287 | 582.28 | 190,246.71 | 33,956.22 | 23,972.26 |
| Rural households (household) | $RH_i$ | 287 | 927 | 7,079,400 | 876,955 | 645,599.50 |
| Slope (degree) | $Sl_i$ | 287 | 0.04 | 11.82 | 2.47 | 2.02 |

Figure 2 shows the spatial variation of the average income; the average living expenditure of these new urban migrants at the prefecture level (Figure 2A,B); the Hukou registration index; and the economic production of secondary and tertiary industries among different prefectures (Figure 2C,D). The incomes and expenditure levels of migrants from rural to urban areas were higher in the economically developed southeastern regions than in the northern regions. The Hukou registration index and the economic production of secondary and tertiary industries were higher in coastal areas, provincial capitals, and economically developed cities.

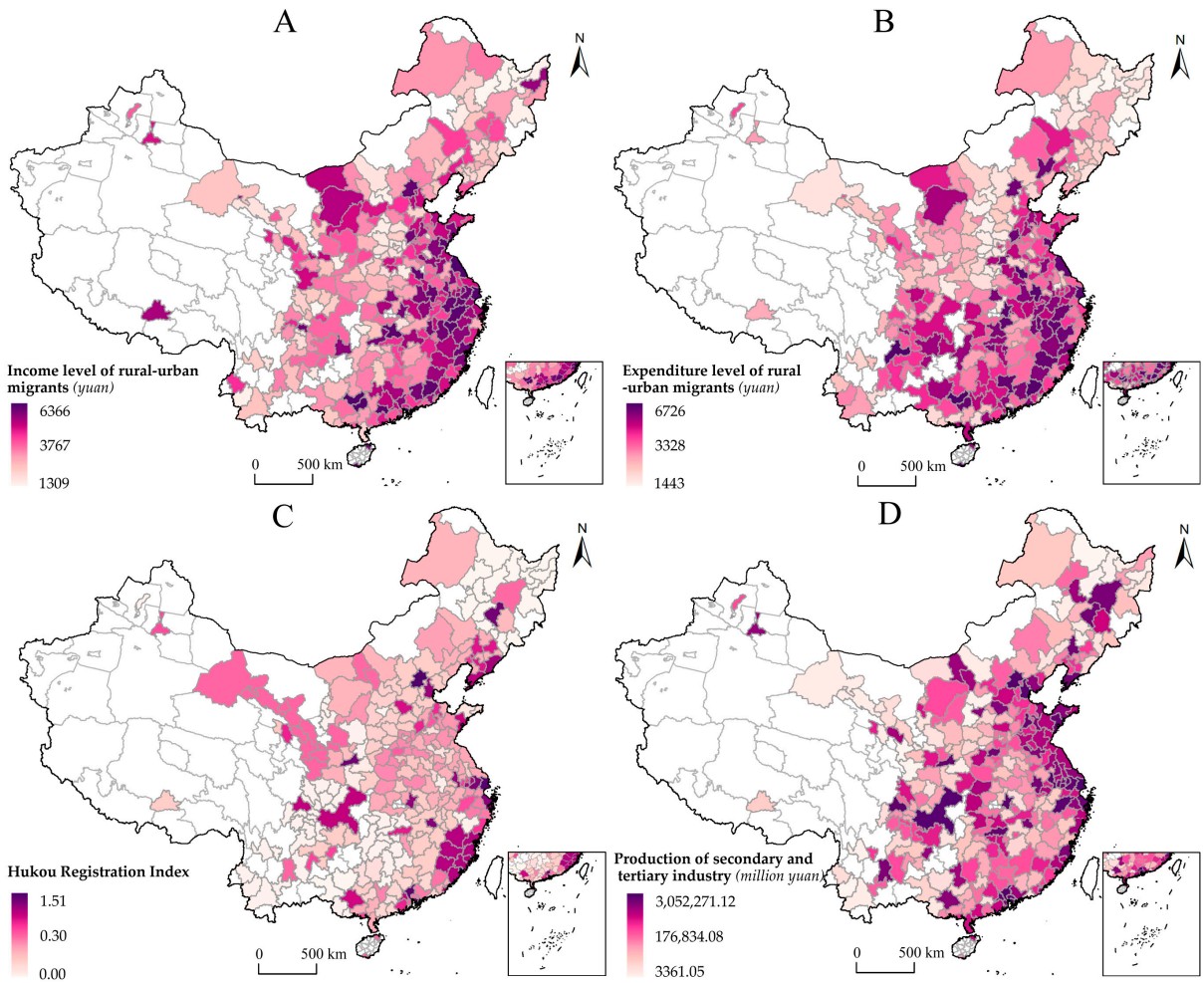

**Figure 2.** (**A**) Average income of rural-urban migrants for each prefecture—2017 data. (**B**) Average living expenditure for rural-urban migrants for each prefecture—2017 data. (**C**) Hukou registration index by prefecture—2016 data. (**D**) Production of secondary and tertiary industries by prefecture—2017 data.

### 2.3. Methodology

2.3.1. The Network/Spatial Lag of X (N/SLX) Model

Traditionally, spatial regressive models were adopted to conduct causal inference with spatial interactions [60]. By introducing a spatial weight matrix, the spatial regressive model augmented the standard linear regression model with an additional spatial lagged term that incorporated the spatial relationship (i.e., neighbourhoods or distance) between observations [61,62]. Spatial regression models have recently been developed as network regression models for applications in non-spatial domains, including political science and public administration [63], where they are used to explore institutional isomorphism or party strategies and economic emphasis [64,65]. The spatial matrixes have been replaced by network matrixes to identify network dependency among entities. Regarding the selection of the specific spatial econometric model, Gibbons and Overman [66] advocated the issues of recognition and causality as the core and proposed a series of basic models, for instance, the spatial lag of X (SLX) model, to identify the externality motivation imposed on X.

Inspired by the existing research, the present study developed a network/spatial lag of X (N/SLX) model to test the significance of the semi-urbanised population's impact on rural construction land. The spatial lag of X (SLX) model, which includes explanatory variables observed on neighbouring cross-sectional units, is the simplest spatial regression model [66]. We developed it by replacing the spatial weight matrix (*W*) with the network

weight matrix (denoted as $^nW$ and calculated in Equation (2)), which was constructed based on the migration estimations among rural and urban regions (i.e., $P_{ij}$ counted in Equation (3)). The general form of the model is expressed as Equation (1):

$$Y_i = \beta_0 + R_i\beta_1 + WU_j\rho + \varepsilon \qquad (1)$$

where the dependent variable $Y_i$ indicates the construction land expansion phenomenon in the rural area $i$; $R_i$ are the local control variables denoting the characteristics of rural areas that might influence the extent of rural construction land, including the number of rural households, average slope, and agricultural output (total production of agriculture, forestry, husbandry, and fisheries) of each prefecture; $U_j$ is the core explanatory variable vector representing the characteristics of the semi-urbanised migrants who lived and worked in the cities (with the subscript $j$ indicating the location of the cities). According to our analysis on the variable selection in this paper, the key explanatory variables include the average income and living expenditure of urban migrants, the production of secondary and tertiary industries (i.e., urban economic development), and the Hukou registration index (i.e., discriminatory policies); $W$ is the migration network matrix generated by Equation (2) (thus, the network lagged term $WU_j$ captures the impact of the semi-urbanised migrants on rural areas via the rural-urban migration network); $\varepsilon$ is the stochastic error term; and $\beta_0$, $\beta_1$, and $\rho$ are the regression coefficient vectors. The dependent variable and all the explanatory variables were logarithmically processed based on the original data to reduce the heteroscedasticity caused by the wide difference in quantity between variables, and the skewed distribution occurred in variables like the extent of rural construction land, the average slope, etc.

If the coefficient of the network lagged variable ($\rho$ values) was significantly different from zero, we would have been able to confirm the hypothesis that the semi-urbanisation of rural-urban migrants was related to the expansion of construction land in rural areas. Since the spatial/network spillover effect is exogenous to the dependent variable, the model can be solved using the ordinary least square (OLS) technique.

2.3.2. Population Migration Network Matrix

In accordance with our hypothesis, the increased extent of rural construction land in home villages can be attributed to the behaviour of semi-urbanised rural-urban migrants, so the spatial difference and direction of this migration should have been measured quantitatively. Unfortunately, neither official statistics nor volunteered geographic information (VGI) data provides sufficient information for this purpose [67]. National population census data of China only present the quantity of immigration in each prefecture; that is, they lack detailed origin information. VGI-derived migration index data (e.g., the Baidu mobility index or the Weibo mobility index) do not cover the entire population and contain mixed population movements for various purposes [6].

Inspired by Liu et al. (2022) [57] and Wang et al. (2019) [68], we employed both VGI data and total immigrants of each prefecture of national population census, and the flow of rural-urban migration between each pair of origin and destination could be inferred. VGI-derived migration index data during the Spring Festival travel season could represent the population movement between rural and urban areas as accurately as possible on one side. The movement of population during this period is dominated by semi-urbanised migrants, which is a unique socio-economic phenomenon in China. It takes place during the Lunar New Year when the aforementioned return to their home villages for family reunions [69]. On the other side, the data of immigrants from the national population census covers every county; thus, the number of immigrants in each city could be aggregated from the counties and be used to revise the VGI-derived migration index data to enhance the authenticity of rural-urban migration. Based on above analysis, we first harnessed VGI data and urban

immigrants of census data to reckon the flow of rural-urban migration between each pair of origin and destination (Equation (2)).

$$W_{ij} = P_{ij} \times IM_j \qquad (2)$$

where $W_{ij}$ represents the migration between the origin rural area $i$ and destination city $j$; $IM_j$ represents the total immigration in city $j$, which is derived from the census data; and $P_{ij}$ is the destination choice matrix that indicates the probability of outmigration from area $i$ to area $j$. We estimated the choice matrix based on Equation (3), where $F_{ij}$ represents the observed inter-regional migration flow obtained from a VGI source.

$$P_{ij} = F_{ij} / \sum_i F_{ij} \qquad (3)$$

Considering the area of rural construction land in this paper is from 2018, we chose the 2018 Weibo mobility index as our data source to construct an inter-regional migration flow matrix $F_{ij}$ [57]. Referring to Lai and Pan (2020) [70], we first obtained the weekly mobility index from 29 January to 18 February (before the Spring Festival) and from February 19 to 12 March 2018 (after the Spring Festival). We then calculated the average flow of people between each pair of origin and destination before the Spring Festival and after the Spring Festival. Specifically, we transposed the population flow matrix before the Spring Festival, given that the population movements before the Festival are dominated by returnees.

Figure 3 shows the top 1000 migration flows in terms of estimated volume, thereby illustrating the major rural-urban migration patterns at the prefecture level. The starting point of the arrow is the origin of migration, and the ending point is the destination, with line thickness and colour representing the population size on each route. Meanwhile, the direction of the arrow is clockwise.

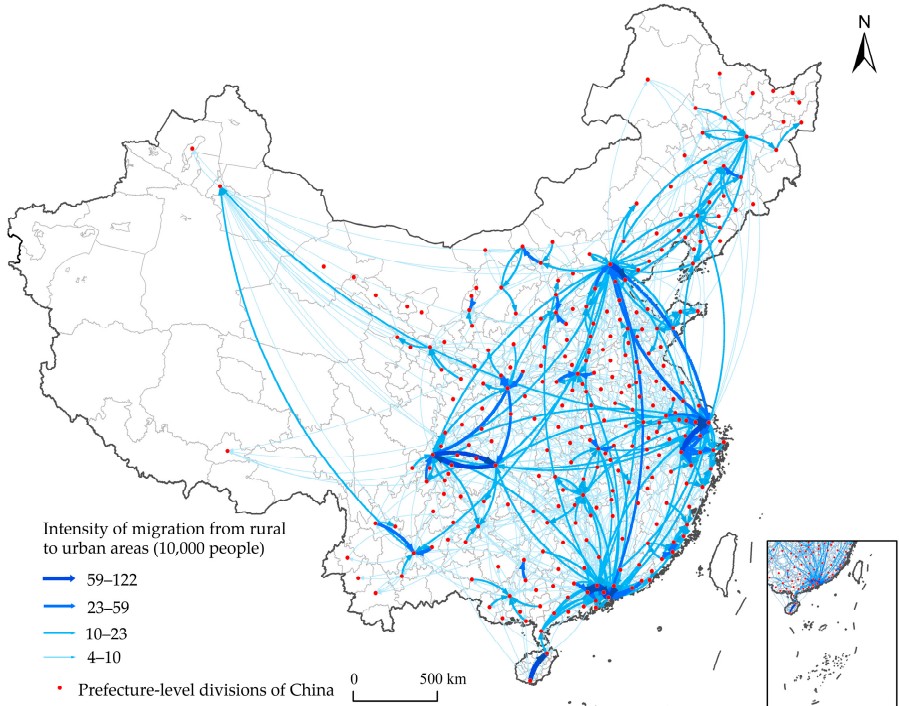

**Figure 3.** Inter-prefecture flow of rural-urban migration—2017 data.

## 3. Results

### 3.1. Regression Results from the N/SLX Model

Because the term *network lagged* was exogenous to the dependent variable, we adopted the OLS technique to estimate the parameters of the N/SLX model. Table 2 displays the results of the network regression model. The (network lagged) incomes of migrants

($W \times {}^{m}Inc_j$) were significantly positively related to the extent of rural construction land, which is in line with the second hypothesis, thereby suggesting that semi-urbanised migrants with higher income levels in cities prominently led to more expansion of rural construction land. The living expenditures of semi-urbanised migrants in urban areas ($W \times {}^{m}Exp_j$) were significantly negatively related to the area of construction land in rural areas, which is consistent with Hypothesis 3. This indicated that the elevated expenses associated with living and housing in urban regions notably decreased the remittances transferred by migrants for investment in rural settlements, hence hindering the expansion of construction land.

**Table 2.** Results from the network/spatial lag of X (N/SLX) model.

| Variables | N | Coefficient | SE | t Statistic | p |
|---|---|---|---|---|---|
| Constant | - | 17.0019 | 3.9145 | 4.343 | 0.0000 *** |
| $\mathrm{Ln}(W \times {}^{m}Inc_j)$ | 287 | 3.2276 | 0.5954 | 5.421 | 0.0000 *** |
| $\mathrm{Ln}\left(W \times {}^{m}Exp_j\right)$ | 287 | −5.3623 | 0.5597 | −9.582 | 0.0000 *** |
| $\mathrm{Ln}\left(W \times HRI_j\right)$ | 287 | 0.5523 | 0.1187 | 4.652 | 0.0000 *** |
| $\mathrm{Ln}\left(W \times STI_j\right)$ | 287 | −0.2415 | 0.1043 | −2.314 | 0.0214 ** |
| $\mathrm{Ln}\left(AFH_i\right)$ | 287 | 0.4512 | 0.0885 | 5.100 | 0.0000 *** |
| $\mathrm{Ln}(RH_i)$ | 287 | 0.2807 | 0.0768 | 3.656 | 0.0003 *** |
| $\mathrm{Ln}(Sl_i)$ | 287 | −0.3783 | 0.0354 | −10.684 | 0.0000 *** |
| Adjusted $R^2$ | | | 0.6142 | | |
| F statistic | | | 66.06 | | |

Note: ***—$p < 0.01$; **—$p < 0.05$.

The (network lagged) household registration index ($W \times HRI_j$) was significantly positively related to the extent of rural construction land, thereby supporting Hypothesis 2, i.e., the high household registration threshold contributed to the expansion of rural construction land through the rural-urban interactions. The extension of rural construction land would grow with the rise of household registration thresholds in cities. The production of secondary and tertiary industry ($STI_j$) was significantly negatively correlated with the extent of rural construction land, which is consistent with the third hypothesis. This suggests that the development of urban economies increased the settlement expectation of the migrants from rural to urban areas, thereby restraining the expansion of rural construction land.

The coefficients of the local control variables aligned with our expectations. Economic development ($AFH_i$) and the number of households ($RH_i$) in the rural area were positively related to the area of construction land, and the geo-morphologic roughness (measured by the average slope $Sl_i$) was negatively related to the area of construction land in the rural areas. Overall, the above results strongly confirm that semi-urbanisation caused the increasing areas in rural construction land through the settlement expectations of semi-urbanised migrants and their remittances from urban to rural areas, thereby supporting Hypothesis 1.

*3.2. Robustness Analysis*

To test the robustness of the baseline N/SLX modelling results, we constructed an alternative model by employing an inverse distance ($d_{ij}^{-1}$)-based spatial weight matrix (${}^{d}W_{ij}$), which consisted of the inverse of distance between prefecture $i$ and prefecture $j$ instead of the migration-based network weight matrix (${}^{m}W_{ij}$) from the baseline model (Equation (1)). Given the similarity of the migration matrix and the distance weight matrix, the coefficient estimates were expected to be similar. The two models are compared in Table 3.

Because the explanatory variables of the two models were the same, the central conclusion remained: there was a significant correlation between the socio-economic status

of semi-urbanised migrants and the extent of construction land in rural areas. However, because the distance decay function did not represent the spatial patterns of migration flows, the overall fitness of the model was reduced (with a slightly lower adjusted $R^2$).

**Table 3.** Robustness test.

| Variables | Baseline Model | Alternative Model |
|---|---|---|
| Constant | 17.0019 *** | 8.5660 *** |
| $\mathrm{Ln}\left(W \times {}^{m}Inc_j\right)$ | 3.2276 *** | 0.4729 * |
| $\mathrm{Ln}\left(W \times {}^{m}Exp_j\right)$ | $-5.3623$ *** | $-1.9095$ *** |
| $\mathrm{Ln}\left(W \times HRI_j\right)$ | 0.5523 *** | 0.1475 *** |
| $\mathrm{Ln}\left(W \times STI_j\right)$ | $-0.2415$ ** | $-0.0339$ |
| $\mathrm{Ln}(AFH_i)$ | 0.4512 *** | 0.5332 *** |
| $\mathrm{Ln}(RH_i)$ | 0.2807 *** | 0.1274 |
| $\mathrm{Ln}(Sl_i)$ | $-0.3783$ *** | $-0.3679$ *** |
| Obs. | 287 | 287 |
| Adjusted $R^2$ | 0.6142 | 0.5881 |
| F statistic | 66.06 | 59.33 |

Note: ***—$p < 0.01$; **—$p < 0.05$; *—$p < 0.1$.

## 4. Discussion

### 4.1. Irrational Non-Localized Demand for Rural Construction Land

This study presents an empirical investigation into the impact of semi-urbanisation among urban migrants on the expansion of rural construction land. Adopting a rural-urban interaction perspective, we developed a migration network-based theoretical framework. Utilizing VGI data and Census data, we examined the pathways through which semi-urbanised migrants influence the expansion of rural construction land. The research findings revealed a significant and irrational non-localized demand for construction land in rural areas, thus contributing significantly to its expansion. This large-scale empirical study provides evidence that corroborates the findings established in a series of smaller-scale, field survey-based studies [52]. In addition, the finding of this study echoes the study of Shi and Wang (2021) [33], who believe there is a relationship between rural outmigration and rural construction land change.

From an individual perspective, investing in homesteads in their home villages can be viewed as a reasonable investment for semi-urbanised migrants, given their comparatively modest expectations of settling down in cities [26]. However, due to the ongoing urbanisation process in China, only a small fraction of rural-urban migrants ultimately return to their home villages after several years of working and living in the cities [54]. This has led to the inefficient use or even abandonment of construction land in rural areas [32]. Considering that a substantial amount of rural construction land primarily serves as a psychological comfort for new urban immigrants facing an uncertain future, this irrational demand for land undoubtedly constitutes a significant waste of resources for the entire society [19].

The urbanisation process, marked by the migration of rural populations to cities, inherently leads to the spatial separation of people and land [25]. China's distinctive land system has given rise to a unique inter-regional interaction network [71], thus establishing a specific channel through which the irrational demands of new immigrants working and living in cities are transmitted to rural areas and resulting in a non-localized demand for construction land. With the further deepening of urbanisation in recent years, small towns close to the rural hinterland have relaxed the restrictions on settlement and housing purchase for non-local residents [36]. This has led many rural-urban migrants working in large cities to choose to acquire real estate in small towns near their home villages [52]. This phenomenon further exacerbates the inefficient expansion of construction land nationwide. These complex network interaction mechanisms also make identifying such inefficient expansion more challenging.

To address these issues, a new research framework is urged to identify the irrational non-localized demand for land under the inter-regional interaction perspective; the combination of VGI data and traditional statistical data has shown its potential to support this research paradigm. In terms of policy practice, more integrated policies for urban and rural development, which take the telecoupled urban and rural land system into account, are needed in the future. More specific policy implications are proposed in the next session.

### 4.2. Policy Implications

Considering the substantial influence of semi-urbanised populations on the expansion of construction land in rural areas, in order to mitigate the inefficient use of rural land, we propose the following policy recommendations.

Firstly, the government should continue to promote reform of the household registration system and lower the barriers to obtaining urban household registration in more cities. The present study has revealed that such discrimination between regular urban residents and new migrants in urban areas led to incomplete urbanisation and the expansion of rural construction land. In recent decades, many cities in China have relaxed the requirements for rural-born migrants to obtain urban household registration, which promoted labour mobility and boosted urbanisation and economic development [36]. Therewith, this study has contributed additional evidence supporting the spillover benefit of improving urban household registration and associated policies, particularly policies in mega-cities. As a result of this study, we found that the household registration index is still high in coastal areas, provincial capitals, and economically developed cities with the most rural-urban migrations.

Secondly, policymakers should extend the coverage of the social security and benefits system, which currently offers limited coverage for the relatively disadvantaged rural migrant workers in urban areas. Disparities persist between new migrants and regular urban residents in the provision of social services (e.g., children's education and healthcare), thereby making family-based migration more difficult. Many new migrants who find employment in the city send a portion of their income back to rural areas to improve the lives of their left-behind families, which ultimately contributes to the expansion of rural construction land. Improving the social security system would not only promote social equity but also help to curb the inefficient expansion of rural construction land.

Thirdly, aside from the policy suggestions aimed at cities, we also provide an alternative suggestion for urbanisation in places where it is easier to settle, such as county towns. County towns are not only equipped with more employment and higher income than rural areas, but also quality public services can be more easily accessed by rural migrants than in mega-cities [72]. Urbanisation based on county towns may afford critical assistance in transforming semi-urbanisation into urbanisation and the restraint of the inefficient expansion of rural construction land. This suggestion is consistent with the guidelines promulgated by the General Office of the Communist Party of China Central Committee and the State Council General Office, thus promoting that urbanisation should pay attention to county towns in all parts of the country.

### 4.3. Limitations

Because no official rural-urban migration statistics were available, we resorted to VGI sources and the Seventh National Population Census dataset to estimate inter-prefecture rural-urban migration flows, which inevitably introduced an element of uncertainty into the study. The concern pertains to the sample coverage of the Weibo mobility index. This index relies on geo-tagged posts from Sina Weibo users (similar to Twitter in Western countries), which can lead to sample bias across different age groups. However, due to the similarity in age structure between new urban migrants and Weibo users [73] and the proven reliability of Sina Weibo data as determined by Liu et al. [57], we believe that such bias had a limited impact. In spite of this, providing more accurate and informative data on migration is imperative for future study.

The ideal explanatory variable to test the hypothesis of the present study would have been the remittances sent by semi-urbanised migrants to their home villages. However, due to advances in modern financial systems, particularly the rise of e-finance, it has become increasingly challenging to trace the destination of money transfers. To circumnavigate this, we introduced an alternative variable called migration network lagged income, that is, remittances sent by new migrants to their home villages after controlling for living and housing expenditures in the cities where they worked [45].

## 5. Conclusions

The present study offers empirical evidence substantiating the hypothesis that the semi-urbanisation of rural migrants has resulted in the expansion of rural construction land in China. Both higher incomes and relatively low settlement expectations in urban areas were positively correlated with the expansion of construction land in rural areas. The negative correlation between living expenditures and the extent of rural construction land also implied a relationship between remittance and rural construction land expansion.

This study demonstrated that advancing the urbanisation process remains an effective means of improving land use efficiency. The development of urban economies not only improves the effective utilization of urban land but also significantly reduces the inefficient use of rural construction land. However, the phenomenon of semi-urbanisation resulting from a lack of complementary urban inclusion policies may substantially diminish the positive impact of urbanisation on enhancing land use efficiency. The findings suggest that, aside from local factors in rural areas, the lifestyles and the socio-economic characteristics of rural-urban migrants contribute to the process of rural construction land expansion.

Yet, it is undeniable that this study did not consider the negative effects of the continuous transformation of semi-urbanisation to urbanisation on the efficiency of rural construction land use. With more and more rural-urban migrations and their gradual integration into cities, large-scale rural construction land may be deserted and become vacant. In this scenario, the improvement of rural construction land utilisation efficiency attributed to the control of low-efficient expansion may be neutralised. In view of urbanisation as an inevitable social trend, exploring the rational reuse of idle rural construction land may be the key for future study.

Additionally, the present study offers an insight into how changes in land use in one area are influenced by the development of other areas far away and mirrors telecoupled land system research findings. Advances in modern transport and communication technologies have lowered the cost of overcoming spatial impedance, so spatial spillover effects have been replaced by a telecoupling process generated by flows of information, people, products, and so on. Future research assessing the impact of spatial policies from a telecoupling system perspective is therefore needed.

**Author Contributions:** Conceptualisation, C.Z.; methodology, Y.W. and C.Z.; software, Y.W.; formal analysis, Y.W.; resources, Y.W., C.Z. and M.Z.; data curation, Y.W. and M.Z.; writing—original draft preparation, Y.W.; writing—review and editing, C.Z. and Y.W.; visualisation, Y.W.; supervision, C.Z.; funding acquisition, C.Z. All authors have read and agreed to the published version of the manuscript.

**Funding:** This study was funded by the Later Stage Program of Philosophy and Social Science Research by the Ministry of Education of the People's Republic of China (Grant number 21JHQ019) and the National Natural Science Foundation of China (Grant number 42371277, Grant number 42371278).

**Data Availability Statement:** Data sources were stated clearly in the section of Data Sources.

**Acknowledgments:** The authors would like to thank the seniors and teachers for their guidance and assistance.

**Conflicts of Interest:** The authors declare no conflicts of interest.

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
