# Peer review of "How Semi-Urbanisation Drives Expansion of Rural Construction Land in China: A Rural-Urban Interaction Perspective"

_land, doi:10.3390/land13010117_

Round 1

Reviewer 1 Report

Comments and Suggestions for Authors

This study utilizes the Weibo Mobility Index and the dynamic monitoring survey data on China's floating population to track migration trajectories and socio-economic characteristics between urban and rural areas. It adopts a network/spatial regression model to analyze the impact of semi-urbanization on rural construction land. Overall, this study is meaningful in its topic selection.

Specific revision suggestions:

1.       Section2, "Theoretical Analysis and Hypothesis," should provide more detailed information to align with the results of the empirical analysis section. The current content is somewhat limited, and it is not clear what specific research hypotheses are being proposed. The research hypotheses should be clearly stated in the text.

 2.       The apostrophe after 'IMj' in Equation (1) should be removed.

 3.       The citation for the "network regression model" in Equation (3) should be indicated in the text with a proper reference.

4.       Section "4.2. Model Specification and Data Sources" primarily provides an explanation of the data. It is redundant to include a separate figure for almost every variable. A textual description and tables would suffice for explaining the data without the need for so many figures.

5.       In the section "5.1. Regression Results From the N/SLX Model," only the reporting and analysis of the results are provided, without delving deeper into the underlying reasons for the positive or negative relationships between variables. Further explanation is needed regarding why certain variables exhibit positive or negative correlations. Additionally, it is unclear on what basis all variables are logarithmically transformed. For example, is it necessary to take the logarithm of variables such as "slope"? The rationale for logarithmic transformation of some variables needs further discussion.

6.       Some of the figure titles in the text are not centered. Additionally, it is unclear why some figures are for the year 2017 while others are for 2018. Furthermore, some figures seem to lack indication of the year altogether.

7.       Section 6, "Conclusion and Policy Implications," should be labeled as 6.2. Additionally, "Conclusion" should be changed to "Conclusions."

8.       The section "Migration Flow from Rural to Urban Areas" being a separate Section seems abrupt. It could be integrated with Section 4 and 5 (as suggested in point 9). Section 3 could be renamed as "Methodology" and further divided into "Data" and "Methods," providing detailed explanations of the data sources and methodology choices. Currently, it is not clear where the data comes from or how the methods were selected.

9.       The current structure of Section 3, "Migration Flow from Rural to Urban Areas," Section 4, "Empirical Study," and Section 5, "Results" appears confusingon.

10.   The innovative aspect mentioned in the paper is the study of the "impact of semi-urbanization on expansion in remote urban areas," but throughout the paper, the research on "remote areas" does not seem to be prominently highlighted.

11.   The "Policy Implications" section presents suggestions based on existing literature, so it is unclear where the author's contributions lie.

Comments on the Quality of English Language

 Minor editing of English language required

Reviewer 2 Report

Comments and Suggestions for Authors

Dear All,

Before the possible publication, a few review questions should be answered:

How does the study contribute to existing literature on the expansion of rural construction land in China?

In what ways does the research address the gap in understanding the impact of semi-urbanization on rural construction land in distant urban areas?

How robust is the methodology used, particularly the combination of Weibo Mobility Index and China Migrants Dynamic Survey data?

Does the network/spatial regression model seem appropriate for analyzing the influence of urban semi-urbanization on rural construction land?

To what extent do the results support the stated hypothesis regarding the correlation between income, household registration, living expenses, city economic development, and the extent of rural construction land?

How well does the study discuss and interpret the positive correlation between the income of rural-urban migrants and the expanding extent of rural construction land?

Are the negative correlations with living expenses of migrants and city economic development adequately explained and justified?

How relevant are the proposed policy implications, particularly the suggestion that urbanization policies play a crucial role alongside rural land use control policies?

Are there alternative policy recommendations that could further address the issue of inefficient expansion of rural construction land?

Please provide essential revisions as soon as possible based on the review questions.

Reviewer 3 Report

Comments and Suggestions for Authors

The reviewed article addresses an essential issue from theoretical and practical perspectives. The text has been carefully prepared based on substantial literature and empirical data. The selection of methods and the reasoning are correct. However, I have two suggestions.

There should be a precise research objective in the abstract. Providing more detailed information about the applied research method is also advisable.

The structure of the study is slightly different from the content of a typical scientific article. Usually, the text is divided into the following sections: Introduction / Literature Review / Research Methodology / Results / Discussions / Conclusions.

Reviewer 4 Report

Comments and Suggestions for Authors

This article analyzed the impact of semi-urbanization on rural land expansion in China using Weibo Mobility Index and migration data. Before publication, the article needs major revisions:

The author proposes that urbanization policies facilitating the settlement and integration of urban migrants into urban life are as crucial as rural land use control policies to curb the inefficient expansion of rural construction land. Fundamentally, this means that as more and more rural people move into cities, the expansion of rural land expansion will naturally decrease. However, balanced development between urban and rural areas may be more conducive to sustainable development. The rural population cannot be reduced blindly, so is there a threshold that is the main reason for the reduction in population size and age structure?

The article may need to further discuss its research limitations and future research directions to enhance its depth and breadth. Moreover, a more in-depth analysis and discussion of the limitations of data sources and the model might be required.

 Minor issues

1.     Please clarify what is the difference between rural areas, semi-urbanization areas and urban areas.

2.     Line 27-28: In China, the rural construction land expanded from 1.25 × 105 km2 in 2000 to 1.44 × 105 km2 in 2020, summarizing the expansion of urban areas in China during that timeframe. Please confirm if this statement is correct, as it appears unreliable in terms of both area and speed.

3.     Figure 1. What is the data source of Rural Population? The article does not seem to mention it.

4.     Line 110. semi-urbanisation should be semi-urbanization.

5.     All data used in the article should clearly indicate the source of the data to increase the credibility of the research.

6.     Line 350. Table 2. 0.0214 is bigger than 0.01.

Reviewer 5 Report

Comments and Suggestions for Authors

This manuscript requires a heightened focus on the logical coherence between the headings, especially concerning the interrelation of elements presented in sections 3 and 4. Additionally, the discussion section should delve into the results with a meticulous emphasis on detail and depth. Given the national scale analysis nature of this study, it becomes paramount to thoroughly explore multiple facets of the country and its geography.

Some specific changes are as follows:

1. Lines 25- 49

It is advisable for the authors to reorganize the content in the first two paragraphs. Authors should either underscore the context of this study or accentuate the phenomenon of rural land expansion in China, despite a concurrent decline in rural population.

2. Line 154

What is the correlation between the data presented in sections 3 and 4 and the corresponding indicators? It appears that these data are not explicitly referenced in section 4.2.

3. Line 193

It is recommended that content such as the data and indicators used be placed in front of the content of section4.2. The study's logical framework should derive from these data and indicators, culminating in a series of results obtained through NRM. Furthermore, in the model usage description (section 4.2), it is recommended to explicitly state that the variables outlined in section 4.2 are incorporated into the model, enhancing reader comprehension.

4. Line 226

I did not find Model Specification in the content of 4.2. It is suggested that 4.2 and 4.1 be switched around to focus on elaborating on the data sources and processing of the study. Exactly how this data is used in the model is elaborated on in the Models section.

5. Lines 267-268

Please describe the estimation process, or the formula.

6. Lines 369-391

It is suggested that the authors revisit and rewrite Section 6. The manuscript extensively employs various data and indicators, and there is potential to delve more deeply and meaningfully into the discussion of the results.

7. Line 392

Why are there two of 6.1? The conclusion section should focus on the results of this study, and the conclusions associated with them.

8. Lines 421-430

This content should be relocated from the conclusion to the discussion section for more appropriateness. For instance, consider incorporating a dedicated section offering practical advice to Chinese policymakers or urban planners on mitigating the ongoing expansion of rural construction land.

Comments on the Quality of English Language

The English expression of this manuscript can be understood.

Round 2

Reviewer 1 Report

Comments and Suggestions for Authors

The authors have addressed my concerns. I recommend accepting it in its current form.

Reviewer 4 Report

Comments and Suggestions for Authors

Since the authors have responded and revised the comments point by point, I think this paper is acceptable for publication.

Reviewer 5 Report

Comments and Suggestions for Authors

The authors carefully revised the manuscript, addressing all of my concerns and making the necessary changes. The paper is now ready for publication in its current form. One small suggestion is that Line 352-357 and Figure 3 would be better placed in the Results section.